# Differences in gene expression may contribute to the racial differences in the risk of MASLD

Ivan P. Gorlov[1], Olga Y. Gorlova[1], Aaron P. Thrift[1,2]*

1 Section of Epidemiology and Population Sciences, Department of Medicine, Baylor College of Medicine, Houston, Texas, United States of America, 2 Dan L Duncan Comprehensive Cancer Center, Baylor College of Medicine, Houston, Texas, United States of America

☙ These authors contributed equally to this work.
* Aaron.Thrift@bcm.edu

## Abstract

Metabolic dysfunction-associated steatotic liver disease (MASLD), formerly known as non-alcoholic fatty liver disease (NAFLD), exhibits marked racial disparities in incidence, progression, and clinical outcomes. While diet, lifestyle, and socioeconomic factors have been shown to influence these disparities, biological mechanisms underlying racial differences in MASLD risk remain poorly understood. We hypothesized that race-associated variation in hepatic gene expression may contribute to differential susceptibility and progression of MASLD. To test this hypothesis, we analyzed publicly available gene expression data from liver biopsies obtained from over 300 Black and White individuals undergoing bariatric surgery. Gene expression profiles were compared across four histological stages: normal liver, MASLD, metabolic dysfunction-associated steatohepatitis (MASH) without fibrosis, and MASH with fibrosis. We identified more than 200 genes that were significantly differentially expressed between Black and White individuals. Genes associated with MASLD progression were significantly enriched among race-specific genes, supporting the hypothesis that racial differences in hepatic gene expression contribute to disease risk and progression. Using histologically normal liver as a reference, we identified race-specific candidate genes potentially driving MASLD progression. These included UCN3 and PRSS3 in Black individuals, and MMP15, LAMB2, LEPR, ELOVL2, CD48, COL5A2, and ICAM1 in White individuals. Notably, divergence in gene expression profiles between racial groups became more pronounced with advancing disease stages, suggesting that race may play an increasingly important role in later phases of MASLD progression. Our findings indicate that differential modulation of hepatic gene expression represents a potential biological mechanism contributing to racial disparities in MASLD. These results highlight the importance of considering race-specific molecular signatures in understanding MASLD pathogenesis and in developing targeted prevention and therapeutic strategies.

**Data availability statement:** Used data can be found in published papers as indicated in the paper.

**Funding:** The author(s) received no specific funding for this work.

**Competing interests:** The authors have declared that no competing interests exist.

## Introduction

Metabolic dysfunction-associated steatotic liver disease (MASLD), previously known as non-alcoholic fatty liver disease (NAFLD), is the most common chronic liver disease in the U.S., affecting up to 100 million Americans [1]. MASLD is a broad term for liver conditions where there is fat accumulation in the liver in people with obesity, insulin resistance, high blood pressure, or elevated cholesterol levels. Up to 75% of overweight people and more than 90% of severely obese people have MASLD [2].

MASLD typically progresses through several stages, including steatosis, which is the initial phase of MASLD, when fat accumulates in the liver; metabolic dysfunction-associated steatohepatitis (MASH), which represents a more advanced phase where inflammation is present in the liver; and fibrosis, which is the development of scar tissue in the liver that can lead to cirrhosis and hepatocellular carcinoma (HCC) [3].

Multiple studies have identified striking racial differences in the prevalence and severity/progression of MASLD between Black and White individuals in the United States. Population-based analyses consistently show that Black adults have a lower prevalence of MASLD compared with non-Hispanic White adults, while White individuals tend to have intermediate prevalence rates between Black and Hispanic populations. Meta-analyses estimate MASLD prevalence at approximately 24–29% among U.S. Hispanics [4], a range that is comparable to that observed in non-Hispanic White populations (25%, [5]). In contrast, the estimated prevalence among Black individuals is substantially lower, at approximately 12% [6]. Although prevalence estimates vary depending on diagnostic modality (e.g., imaging-based assessments versus serum liver enzymes) and the populations studied, there is broad agreement that MASLD burden is highest among Hispanics and Whites and lower among Black individuals, even after adjustment for demographic and metabolic risk factors [5]. Notably, African Americans demonstrate a lower prevalence of hepatic steatosis and MASLD despite having a similar or greater burden of cardiometabolic risk factors compared with White individuals.

Racial differences are observed not only in MASLD risk but also in disease etiology and severity. African Americans with MASLD tend to exhibit milder histopathologic features, including lower degrees of steatosis and reduced rates of advanced fibrosis and cirrhosis, compared with White patients [7]. Consistent with these findings, a large cohort study reported that White patients had significantly higher prevalence of nonalcoholic steatohepatitis (NASH) and advanced fibrosis, whereas African American patients had markedly lower rates of stage 3–4 fibrosis [8]. Despite the generally lower disease severity observed in Black individuals, racial differences in MASLD-related mortality do not consistently favor one group, highlighting the need for further investigation into the mechanisms underlying racial disparities in MASLD progression and outcomes [9]. Samji et al. [10] analyzed racial disparities in diagnosis and prognosis of MASLD and concluded that "studies are urgently needed to identify determinants of MASLD disparities and design appropriate intervention strategies to reduce racial disparities to improve MASLD-related morbidity and mortality".

We hypothesized that racial differences in prevalence and progression of MASLD are at least partially related to racial differences in gene expression profiles in liver.

To test the hypothesis we reanalyzed liver gene expression data generated by Subudhi et al. [11]. The goal of the original study was to identify hepatic gene expression patterns associated with patterns of liver injury in a high-risk cohort of adults with obesity. The study provides information on ethnicity, age and gender for all study participants. We reanalyzed gene expression data focusing on comparisons between Black and White individuals in the context of the MASLD risk and progression.

## Methods

### Data

We performed a secondary analysis of gene expression data generated by Subudhi et al. [11], downloaded from the gene expression omnibus database GSE: https://www.ncbi.nlm.nih.gov/geo/query/acc.cgi?acc=GSE163211. Gene expression data of liver biopsies from 98 obese Black and 211 obese White individuals were available. The dataset also provided gender, age, and body mass index (BMI) for each study participant. Gene expression of 800 genes was analyzed. These genes were selected based on published evidence supporting their roles in liver disease and fibrosis, reflecting a candidate gene approach [11–13]. To account for the preselected nature of this gene set and to minimize potential selection bias, we used these 800 genes rather than the entire human genome as the background for gene set enrichment analysis. Expression was estimated using NanoString nCounter. Raw data were normalized to five housekeeping genes *CLTC*, *GUSB*, *PGK1*, *SDHA*, and *TUBB* known to be ubiquitously expressed across different tissues and conditions.

### Statistical analysis

Study subjects were stratified into Black and White groups based on self-reported race. The subjects were further stratified based on stage of progression: (1) histologically normal liver from obese patients, (2) steatosis, (3) MASH without fibrosis, and (4) MASH with fibrosis. Student's t-test was used to compare mean expression between groups. False discovery rate (FDR) was used to control for multiple testing: findings with FDR < 0.1 were considered statistically significant. We used FDR threshold of 0.1 because our analysis is exploratory and aims to identify and prioritize genes with potentially meaningful racial differences in MASLD risk and progression.

To identify genes associated with MASLD progression, patients with histologically normal liver were denoted as "Group 1", patients with steatotic liver as "Group 2", patients with MASH without fibrosis as "Group 3", and patients with MASH with fibrosis as "Group 4". We expected that the expression level of a gene associated with MASLD progression will show a significant (positive or negative) correlation with the progression stage. We used Spearman's rank correlation coefficient $r_s$ to quantify the association. Rank correlation analysis was supplemented by ordinal regression (proportional odds model) to account for the ordered nature of consecutive MASLD stages. We also performed a pairwise comparison of mean expression in steatotic liver, liver of patients with MASH without and with fibrosis using the gene expression in histologically normal liver as a reference. Two-way analysis of variance (ANOVA) was used, with race and progression stage as categorical predictors and gene expression level as a continuous dependent variable, to jointly estimate effects of race and stage on gene expression level. Interaction term was included in the model. The analysis was conducted in R using "aov" command.

For gene enrichment analysis we used the Database for Annotation, Visualization and Integrated Discovery (DAVID) [14]. Only genes with statistically significant differences in expression level after adjustment for multiple testing were used in the gene enrichment analysis.

## Results

### Epidemiological characteristics of Black and White individuals

Fig 1 illustrates the distribution of Black and White individuals by age (left panel) and BMI (right panel). No significant differences were observed between Black and White participants for either age or BMI. The average age in White individuals

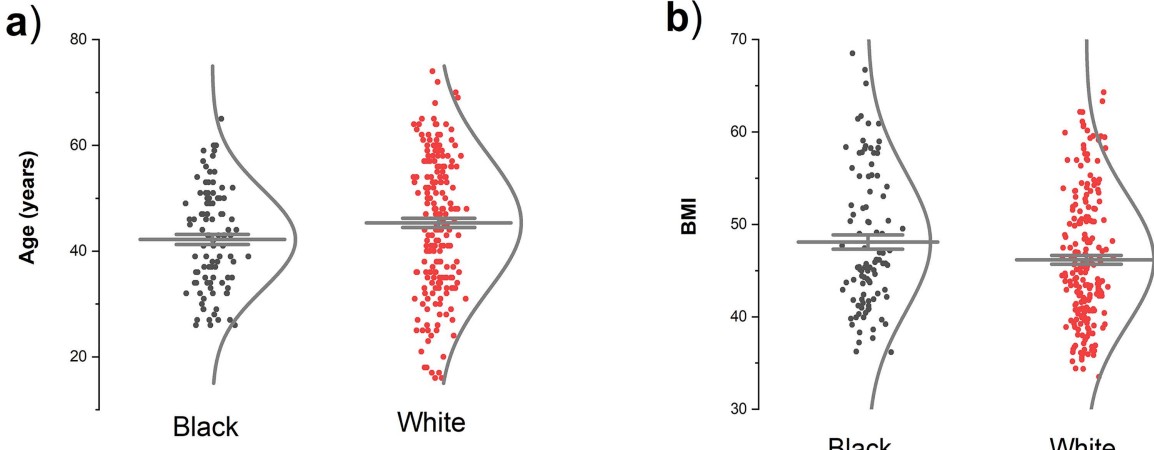

**Fig 1. Age (a) and BMI (b) differences between Black and White individuals.** Long horizontal line indicates mean and two short lines standard errors of means (SE).

was 45.2 ± 1.1 and in Black individuals 42.4 ± 1.5 years (t = 1.6, p = 0.09). BMI in Black individuals was 48.0 ± 1.1 and in White individuals it was 46.2 ± 0.7 kg/m² (t = 1.2, p = 0.25).

Although the compared groups did not differ by age or BMI, we were unable to compare them with respect to clinical comorbidities due to the lack of individual-level data. This limitation should be considered when interpreting the results of this study.

## Genes differentially expressed between Blacks and Whites across stages

We identified genes differentially expressed in liver of Black and White individuals by comparison of the mean expression according to (1) histologically normal liver, (2) liver with steatosis, (3) liver of MASH patients without fibrosis, and (4) liver of MASH patients with fibrosis (Table 1). The complete list of genes differentially expressed (FDR < 0.1) can be found in S1 Table. Differentially expressed (DE) genes identified in histologically normal liver tissue (Group 1) can be considered candidate genes that may contribute to racial differences in MASLD risk. In contrast, DE genes detected in Groups 2–4 likely represent genes associated with MASLD progression.

The number of DE genes increased with disease severity, with 41 genes identified in Group 1, 141 in Group 2, 316 in Group 3, and 136 in Group 4. This pattern indicates that racial differences in hepatic gene expression between Black and White individuals expand as MASLD progresses. The smaller number of DE genes observed in Group 4 compared with Group 3 is likely attributable to the markedly unbalanced sample size in Group 4 (68 White vs. 13 Black individuals). Because statistical power is largely driven by the smaller group, this imbalance results in wider confidence intervals and

**Table 1. Number of genes differentially expressed between Black and White individuals for different stages of liver disease.**

| Stage | Number of DE genes | Sample size (Blacks) | Sample size (Whites) |
|---|---|---|---|
| Normal liver (Group 1) | 41 | 36 | 36 |
| Steatotic liver (Group 2) | 141 | 27 | 58 |
| MASH without fibrosis (Group 3) | 316 | 22 | 49 |
| MASH with fibrosis (Group 4) | 136 | 13 | 68 |

reduced sensitivity to detect true expression differences. In addition, smaller sample sizes are more susceptible to outliers and random variability, further limiting detection power. Consequently, direct comparison of p-values across groups with balanced and unbalanced sample sizes may be misleading.

We conducted permutation analyses to evaluate the impact of sample size imbalance. Specifically, in Group 3 ("NASH without fibrosis"), we performed permutation analyses by randomly selecting 13 out of 22 Black individuals and comparing them with 49 White individuals. Across 50 permutations, the average number of DE genes was 108.5±4.3, substantially lower than the 316 DE genes identified in the full dataset.

Analysis of mean absolute differences in gene expression between Black and White individuals across consecutive MASLD stages further supports the conclusion that Black versus White divergence in gene expression profiles increases with disease progression. The mean absolute log ratios of gene expression between Black and White individuals were 0.103±0.005 for normal liver histology, 0.121±0.008 for steatotic liver, 0.133±0.005 for NASH without fibrosis, and 0.161±0.007 for NASH with fibrosis.

To address this limitation, we performed a permutation-based analysis to account for differences in sample size across disease stages. We randomly selected 13 Black and 13 White individuals from each group and compared their mean expressions. Fifty permutations per stage were conducted. The number of significant findings (FDR<0.1) was 7.2±0.9 for Group 1, 10.6±0.7 for Group 2, 28.5±0.8 for Group 3, and 34.9±1.1 for Group 4, providing a clear indication that the number of significant differences between Black and White individuals in gene expression increases with MASLD progression. The results are further supported by an analysis of absolute differences in gene expression across stages. We computed absolute LOG ratio of mean expression in Black and White individuals across the genes. This was done separately for the four groups. We found a significant positive correlation between the absolute LOG ratio and the stage of progression: $r=0.24$, $N=3,200$, $p=2.3 \times 10^{-11}$.

As illustration of the differences in gene expression between Black and White individuals, Fig 2 shows top genes up- (upper panels) and downregulated in Black compared to White individuals in "all stages together" analysis.

## Genes differentially expressed in stage transition

Three consecutive transitions associated with MASLD progression were analyzed: (1) the transition from a liver with normal histology to steatotic liver, (2) the transition from steatotic liver to MASH without fibrosis, and (3) the transition from MASH without fibrosis to MASH with fibrosis. We wanted to identify genes differentially expressed between MASLD consecutive stages. We ran the pooled analysis with both races together, as well as race-stratified analyses. In the analysis of both races together, no differentially expressed genes were detected for the first transition. For the second – steatotic liver to MASH without fibrosis transition, five differentially expressed genes were detected, with 4 of them (*PON3, RPS13, EIF3E*, and *HRG*) downregulated and *SERPINE1* upregulated in MASH without fibrosis compared to steatotic liver. In the transition from MASH without fibrosis to MASH with fibrosis, seven genes were found to be differentially expressed: *IFI30, SPECC1L, ATP6V1F, COL5A1, SERPINC1, LGALS3,* and *COL3A1*. All of them except *SERPINC1* were upregulated in fibrotic MASH.

In the race-stratified analysis no differentially expressed gene was found for the transition from normal histology to steatotic liver. In the transition from steatosis to "MASH without fibrosis", *PCK1* gene was significantly downregulated in Black individuals. Three genes: *PON3, RPS13,* and *EIF3E* were significantly downregulated in MASH without fibrosis compared to steatotic liver in White individuals. For the transition from MASH without fibrosis to MASH with fibrosis, *UCN3* gene was downregulated in fibrotic liver in Black individuals. In White individuals, the only differentially expressed gene in fibrosis free versus fibrotic MASH was *IFI30* which was significantly upregulated in fibrotic liver of White individuals. S2 Table provides statistics for the stratified and all-together analyses.

## Genes associated with MASLD progression

We considered histologically normal liver, steatosis, MASH without and MASH with fibrosis as consecutive stages of MASLD progression and wanted to identify genes associated with MASLD progression. To identify genes associated with

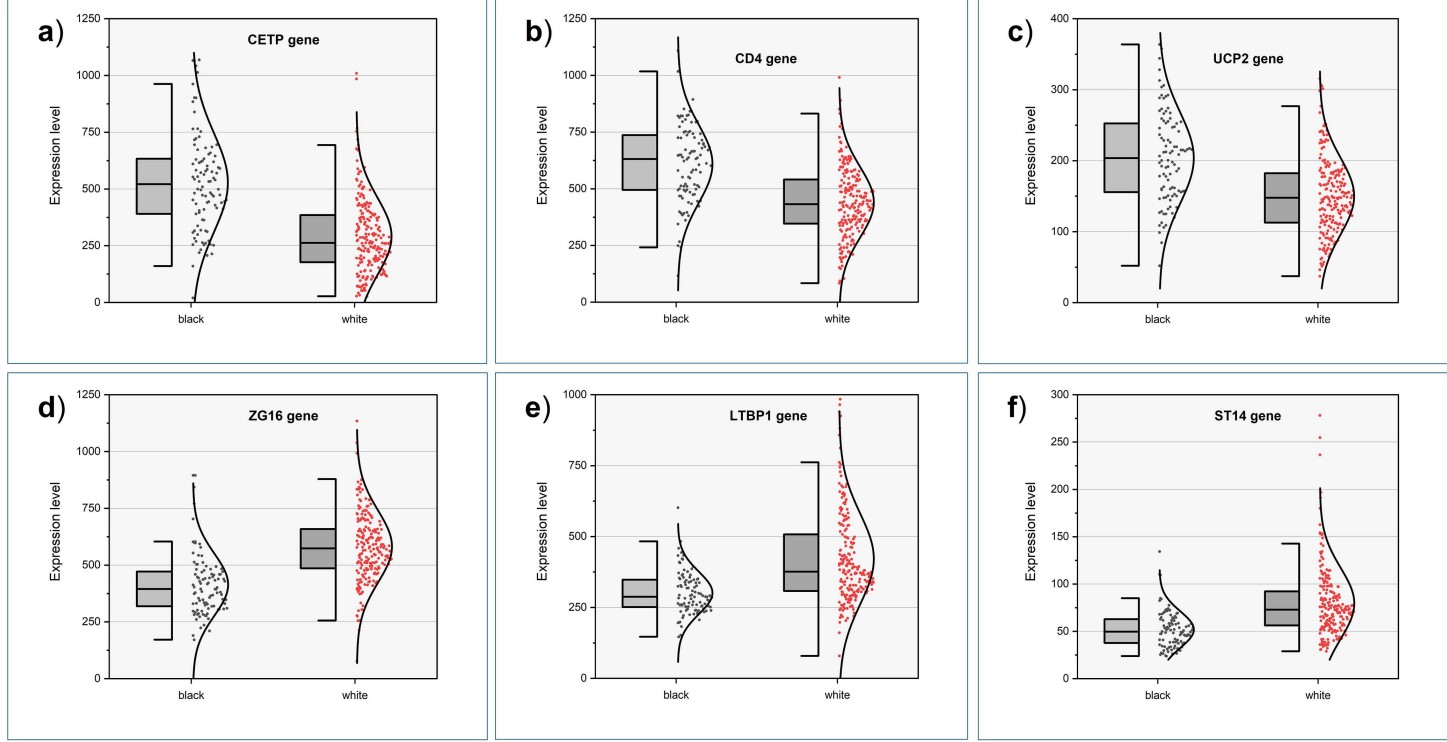

**Fig 2. Top up- (panel a-c) and downregulated (panels d-f) genes in liver samples derived from Black compared to White individuals.** The bracket line indicates 95th and square 75th percentiles. Horizontal line indicates the means.

MASLD progression, for each gene we have computed rank correlation between consecutive stages of MASLD progression and the gene expression level. In the joint analysis, 28 genes were negatively and 59 genes were positively associated with progression. The list of significant genes can be found in S3 Table. Fig 3 shows 3 top genes negatively (upper panel) and positively (lower panel) associated with progression. In analyses stratified by race we have identified 3 genes, *EIF3L*, *HAAO*, and *PPP1R1A*, that were negatively associated with progression in Black individuals. Interestingly, these 3 genes were race-specific and did not show an association in White individuals or in the all-together analysis. In White individuals, 22 genes were negatively and 41 positively associated with MASLD progression.

We also performed ordinal regression analyses using the combined sample (Black and White individuals together) as well as race-stratified analyses. The results are presented in three additional sheets in S3 Table. In total, 76 genes were identified as significantly associated with disease stage in at least one analysis. The majority of these genes (73 of 76) were also detected using Spearman's rank correlation. Three genes: CFB, PKLR, and UBL5 were uniquely identified by ordinal regression but not by correlation analysis. Overall, the ordinal regression results are consistent with and complementary to those obtained using rank correlation, while also identifying additional genes whose expression levels are associated with MASLD progression.

## Genes associated with MASLD progression are enriched by the genes differentially expressed between Black and White individuals

We examined the overlap among three groups of genes: (i) genes differentially expressed between consecutive MASLD stages, (ii) genes whose expression levels showed a significant correlation with MASLD stages coded as 1–4, and (iii)

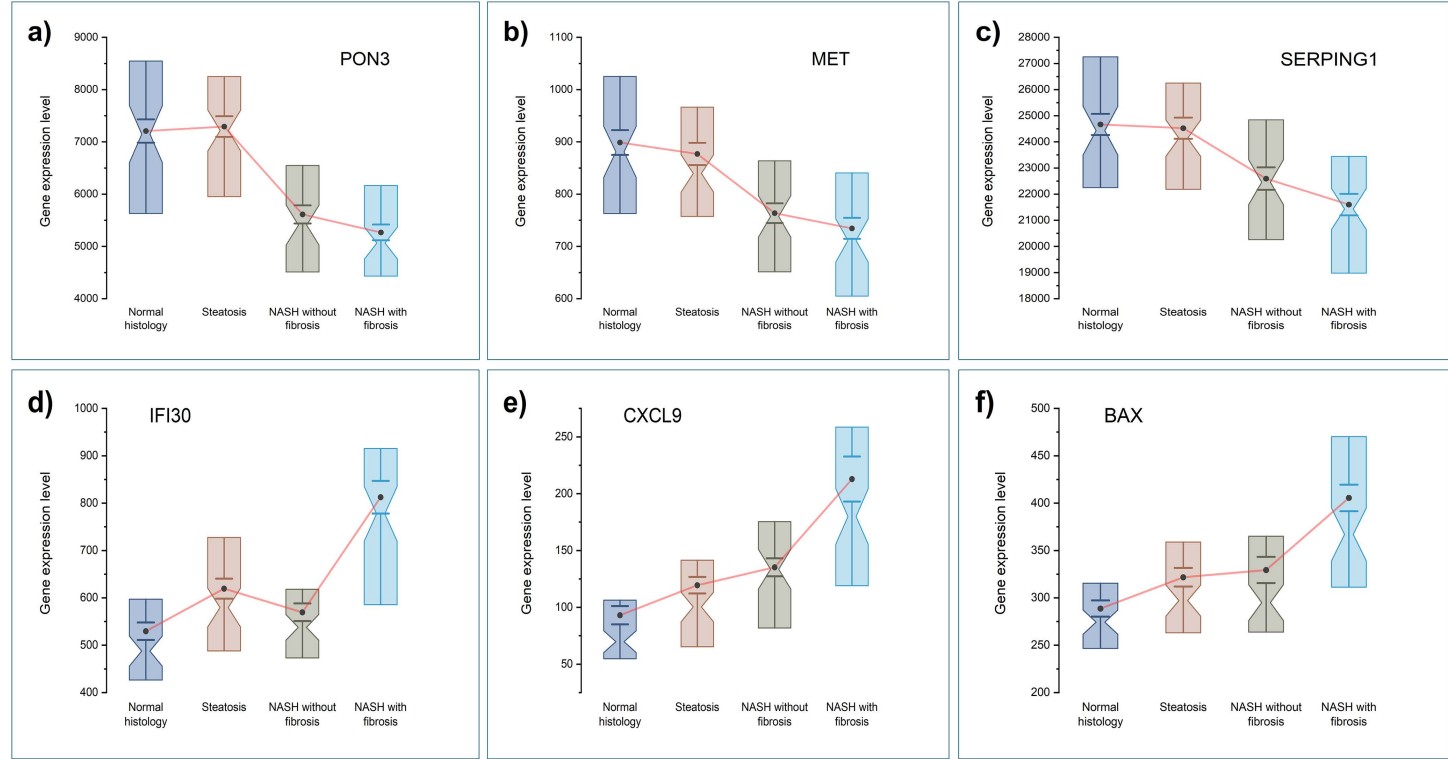

**Fig 3. Top genes negatively (a-c) and positively (d-f) associated with MASLD progression.** Black dots indicate means, whiskers standard errors of means, notched boxes indicate 50% of the data with position of notch indicating median.

genes differentially expressed between Black and White individuals at any of the following stages: (1) histologically normal liver, (2) liver with steatosis, (3) liver from MASH patients without fibrosis, and (4) liver from MASH patients with fibrosis. For each category we included all genes detected in all together, stage specific or race specific analyses. We found that genes associated with stage transition (29 genes) or MASLD progression (93 genes) are enriched by race specific genes (genes that show any association in only one race) (Fig 4). The proportion of genes differentially expressed between Black and White individuals among the genes associated with MASLD progression was $40/93 = 0.43$ and among genes associated with stage transition $20/29 = 0.70$. The expected proportions computed as a product of corresponding frequencies are 0.04 for overlapping race with progression and 0.01 for overlapping race with stage transition. Both differences are statistically significant.

## Joint analysis of race and stage effects on gene expression

We used a two-way ANOVA to estimate the effects of race and stage (categorical predictor variables) on gene expression levels (continuous dependent variable), including a race×stage interaction term in the model. We identified 232 genes for which the main effect of race remained statistically significant in the presence of the race x stage interaction term after adjustment for multiple testing ($p < 0.05$). In contrast, the main effect of stage remained statistically significant for only 56 genes after multiple testing correction. Twenty-six genes were significantly associated with both race and stage, which is comparable to the expected number (16.2) of genes significant for both factors under independence (without the interaction term). The results of this analysis are presented in S4 Table.

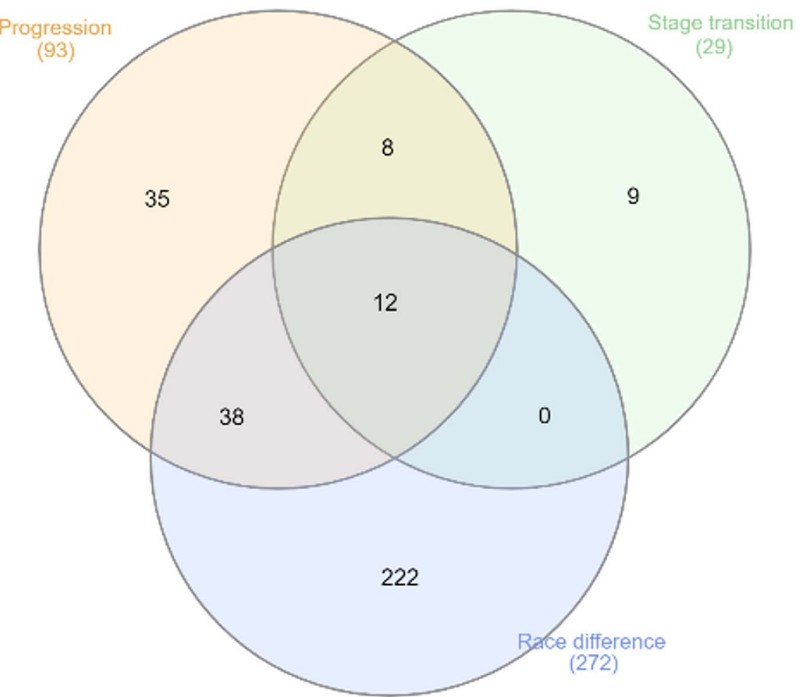

**Fig 4. Venn diagram of the gene associated stage transition (29 genes), MASLD progression (93 genes), and the genes differentially expressed between Black and White individuals (272 genes).**

We did not observe any statistically significant interaction "race x stage" term after adjustment for multiple testing. The most significant interaction before adjustment for multiple testing was observed for *VPS41* gene (p = 0.0003). Fig 5 shows expression of *VPS41* in study participants stratified by MASLD stage and race. The difference between MASH with fibrosis in Black individuals – "F_B" and MASH with fibrosis in White individuals – "F_W" are statistically significant (p = 0.00004).

## Race-specific MASLD-associated genes

We aimed to identify MASLD-associated genes that remained significant after adjusting for the total number of genes and statistical tests performed. This analysis revealed nine genes associated with MASLD in only one racial group, representing race-specific MASLD-associated genes. Here is the list of the genes in alphabetic order:

1. Collagen Type V Alpha 2 Chain (**COL5A2**) was positively associated with MASLD progression in Whites: ρ = 0.28. N = 211, p = 0.00003. The gene does not show correlation MASLD progression in Blacks: r = 0.01. N = 98, p = 0.83.

2. C-X-C Motif Chemokine Ligand 9 (**CXCL9**) was upregulated in steatotic liver compared to the liver of normal histology in Whites 71.7 ± 6.7 versus 125.6 ± 8.3, t-test = 5.1, p = 0.00001. In Blacks expression of CXCL9 was very similar in normal histology and steatotic liver: 114.4 ± 10.1 versus 106.3 ± 11.2.

3. Expression of Eukaryotic Translation Initiation Factor 3 Subunit L (**EIF3L**) negatively correlated with MASLD progression in Blacks: r = −0.41. N = 98, p = 0.00003. The gene doesn't show significant correlation with MASLD progression in Whites: r = −0.09. N = 211, p = 0.21.

4. Elongation of Very Long Chain Fatty Acids-Like 2 (**ELOVL2**) shows significant positive association with MASLD progression in Whites: r = 0.28. N = 211, p = 0.00003, while correlation in Blacks was not statistically significant: r = 0.14. N = 98, p = 0.16.

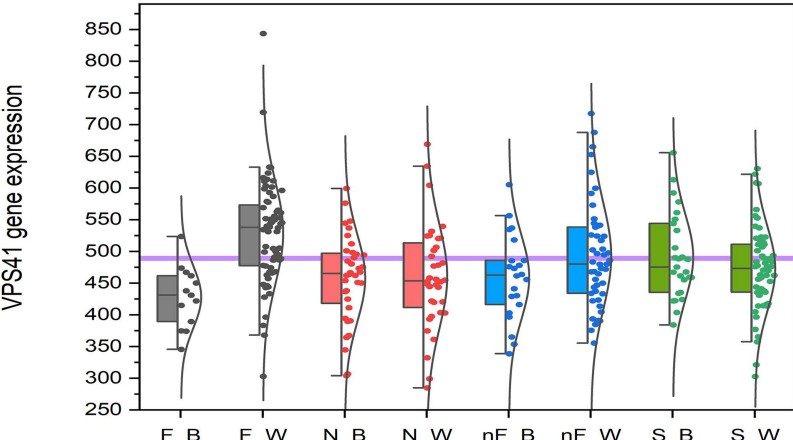

**Fig 5. Stratified analysis of the expression of VPS41 gene.** F – MASH with fibrosis, N – histologically normal liver, nF – MASH without fibrosis, S – steatosis, B – Black individuals, W – White individuals. Horizontal line shows overall average.

5. 3-Hydroxyanthranilate 3,4-dioxygenase (**HAAO**) negatively associated with MASLD progression in Blacks: r = −0.42. N = 98, p = 0.00001 but not in Whites: r = −0.1. N = 211, p = 0.13.

6. Intercellular Adhesion Molecule 1 (**ICAM1**) positively associated with MASLD progression in Whites: r = 0.30. N = 211, p = 0.000008, while correlation in Blacks was not statistically significant: ρ = 0.18. N = 98, p = 0.07.

7. Interferon Gamma Receptor 2 (**IFNGR2**) also shows positive association with MASLD progression in Whites: r = 0.28. N = 211, p = 0.00003 but not in Blacks: r = 0.12. N = 98, p = 0.22.

8. Phosphoenolpyruvate carboxykinase 1 (**PCK1**) is significantly downregulated Blacks in transition from steatotic liver to NASH without fibrosis: 12710 ± 1240 versus 23383 ± 1633, t-test = 4.7, p = 0.00002. In Whites the difference was not significant: t-test = 0.6, p = 0.57.

9. Protein phosphatase 1 regulatory inhibitor subunit 1A (**PPP1R1A**) gene's expression level negatively associated with MASLD progression in Blacks: r = 0.30. N = 98, p = 0.000003, while correlation in Whites was not significant: r = −0.13. N = 211, p = 0.06.

## Gene network analysis of race-specific MASLD-associated genes

We used STRING [15] to build protein-protein association network between identified MASLD-related race-specific genes. We found that *IFNGR2* interacts with *CXCL9* based on analysis of co-expression, experimentally determined protein/protein interactions, and text mining. CXCL9 was predicted to interact with *ICAM1* based on co-expression and text mining (Fig 6).

## Gene set enrichment analysis

We assessed functional enrichment for three groups of the genes: (i) Genes differentially expressed between consecutive stages – stage transition genes, (ii) Genes whose expression positively or negatively correlates with consecutive stages numbered from 1 to 4, and (iii) Genes differentially expressed between Black and White individuals across all stages together or on any given stage. We used all 800 genes as a background to account for the fact that the genes were pre-selected. Genes from the first category were enriched by genes associated with Ehlers-Danlos Syndrome (EDS) (p-value

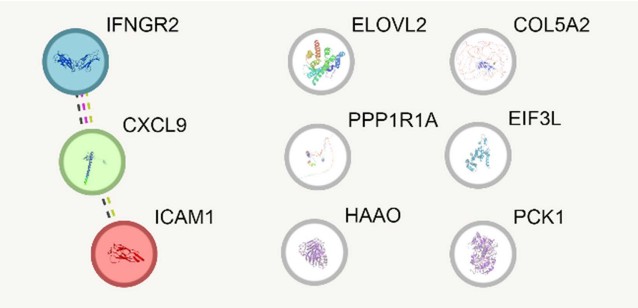

**Fig 6. STRING-generated gene-gene interaction network for 9 MASLD-related race-specific genes.** Black dotted line indicates interaction based on analysis of co-expression, pink dotted line – experimentally identified protein/protein interactions, and green dotted line – gene-gene interaction predicted based on text mining.

adjusted for multiple testing = 0.03). EDS comprises a group of inherited connective tissue disorders characterized by joint hypermobility, skin hyperextensibility, and tissue fragility [16]. Genes associated with MASLD progression are significantly enriched by extracellular exosome genes (p = 0.0004), collagens (p = 0.0006), and the genes associated with endoplasmic reticulum (p = 0.0008). Genes differentially expressed in livers from Black and White individuals do not show any enrichment.

## Discussion

We found extensive race differences in gene expression profiles in the liver. Two hundred thirty-five genes, or 29% of 800 analyzed genes were significantly up- or downregulated in Black compared to White individuals. The number of genes differentially expressed between the races was higher than the number of the genes differentially expressed in stage transition of MASLD progression in "all together" analysis: after adjustment for multiple testing only 99 genes remained significantly associated with MASLD in all together or race stratified analyses. Similar results were obtained using two-way ANOVA: 232 genes were significantly associated with race while only 56 genes were significantly associated with stage (S4 Table). MASLD associated genes are enriched by the genes differentially expressed between Black and White individuals: out of 99 genes associated with stage transition or MASLD progression, 51 (52%) were differentially expressed between Black and White individuals. Similar enrichment was observed for the ANOVA-detected differentially expressed genes: out of 56 genes associated with stage, 27 (48%) were differentially expressed between Black and White individuals. The overlap between MASLD-associated and race specific genes strongly suggests that the genes differentially expressed between Blacks and White individuals play a role in MASLD development and progression and contribute to MASLD racial disparities.

The results of our study demonstrate that stratification of study subjects by race can lead to a better understanding of MASLD pathogenesis. Using histologically normal liver as a reference group, we have identified a number of race-specific DE genes. *UCN3* and *PRSS3* were significantly upregulated in MASH with fibrosis compared to histologically normal liver. The genes were not detected in "all together" or "White individuals only" analyses despite larger sample sizes. *MMP15* gene is differentially expressed between normal liver and MASH with liver in White individuals only and not detected in "all together" or "Black individuals only" analyses. The same is true for *LAMB2, LEPR, ELOVL2, CD48, COL5A2,* and *ICAM1* genes. Thus we have identified 7 genes for MASH with fibrosis specific to White individuals. Therefore, our analysis for the first time identified candidate genes for race-specific drivers of MASLD progression. One of the most intriguing findings of this analysis is the observation that gene expression between Black and White individuals diverge with MASLD progression. The observation is supported by a higher number of differentially expressed genes at more advanced stages,

and by the increase in absolute difference between Black and White individuals in expression associated with MASLD progression. This means that taking into account race may be especially important for analysis of advanced stages of MASLD.

Widespread differences in gene expression profiles between White and Black individuals can be due to dietary and genetic differences between the races. Diet is known to alter gene expression in liver through DNA methylation and histone modifications [17]. There are known differences between Black and White individuals in dietary preferences and eating patterns [18]. Black men and women reported lower intakes of vegetables, potassium, and calcium, and more saturated fat compared to White individuals [19]. Race and geographic region have been shown independently and synergistically influencing dietary intakes [20]. It is possible, therefore, that dietary differences between Black and White individuals contribute to the profound race differences in gene expression profiles observed in this study. Genetic differences between races may also contribute to the differences in gene expression. About 15% of single nucleotide polymorphisms (SNPs) are race specific [21]. A number of MASLD-associated SNPs identified by genome wide association studies (GWAS) are race specific [22]. Since many causal GWAS detected SNPs influence gene expression level [23,24] the genetic differences between Black and White individuals may influence MASLD incidence and progression by modulation of the gene expression profiles in liver.

A major conclusion of this study is that racial differences in gene expression are more pronounced in diseased liver compared with histologically normal liver. Specifically, the number of nominally significant differentially expressed (DE) genes increased from 41 in normal liver to 141 in steatotic liver, 315 in NASH without fibrosis, and 136 in fibrotic NASH. Notably, all genes that remained significant after adjustment for multiple testing were those differentially expressed between Black and White individuals in diseased liver. These nine genes provide strong evidence for their role in racial differences in MASLD progression and represent potential candidate targets for MASLD treatment and management.

Here, we provide a description of the race-specific MASLD-related genes that remained statistically significant after adjustment for multiple testing. Collagen type V alpha 2 chain (COL5A2) encodes the α2 chain of type V collagen, a component of the extracellular matrix (ECM) that co-assembles with type I collagen and regulates fibrillogenesis and matrix organization in various tissues, including the liver's interstitial matrix. Type V collagen has been consistently implicated in tissue fibrosis across multiple organs. Moreover, increases in type V collagen levels and neoepitopes have been associated with advancing hepatic fibrosis in experimental models. These findings suggest a direct link between COL5A2-mediated ECM remodeling and the fibrogenic response during chronic liver injury [25].

C-X-C motif chemokine ligand 9 (CXCL9) is an interferon-inducible chemokine that plays a central role in immune cell recruitment and inflammatory signaling in the liver, particularly under metabolic stress and steatohepatitis. Transcriptomic studies of human NAFLD/MASLD tissue show that CXCL9 expression increases during the transition from simple steatosis to inflammatory NASH and correlates with activation of pro-inflammatory pathways, suggesting a role in disease severity [26]. In murine MASH models, CXCL9 disrupts the balance between regulatory T cells (Tregs) and Th17 cells, promoting pro-inflammatory Th17 responses that exacerbate liver injury [27]. Moreover, CXCL9 signaling through its receptor CXCR3 may influence hepatic stellate cell activation and immune cell infiltration, processes that drive fibrogenesis, neoangiogenesis, and chronic inflammation—hallmarks of MASLD progression [28].

Eukaryotic translation initiation factor 3 subunit L (EIF3L) is a component of the eIF3 complex, a central regulator of cap-dependent mRNA translation that integrates cellular stress, nutrient availability, and inflammatory signaling—processes closely linked to MASLD. Alterations in eIF3 activity have been shown to modulate metabolic pathways, endoplasmic reticulum stress responses, and inflammatory signaling cascades that contribute to MASLD progression [29,30].

Elongation of very long-chain fatty acids protein 2 (ELOVL2) is a key enzyme involved in the elongation of polyunsaturated fatty acids and plays a critical role in regulating lipid homeostasis, inflammation, and fibrogenesis—processes central to MASLD development and progression. Both animal and human studies show that reduced ELOVL2 activity or

expression alters hepatic fatty acid profiles, promoting lipid accumulation, oxidative stress, and pro-inflammatory signaling in the liver [31,32].

3-Hydroxyanthranilate 3,4-dioxygenase (HAAO) encodes a key enzyme in the kynurenine pathway of tryptophan metabolism. Dysregulation of kynurenine pathway enzymes, including HAAO, has been linked to increased production of reactive oxygen species and pro-inflammatory metabolites, which can exacerbate hepatocellular injury and promote fibrogenesis [33,34]. Experimental studies show that altered flux through the HAAO-dependent branch of tryptophan metabolism affects macrophage activation, inflammatory cytokine signaling, and tissue fibrosis in chronic inflammatory states, including liver disease [35]. Importantly, population studies have reported racial differences in circulating tryptophan metabolites between Black and White individuals in the United States, independent of traditional metabolic risk factors, suggesting that variation in kynurenine pathway regulation may contribute to disparities in MASLD severity and progression [36].

Emerging evidence suggests that genetic variation in ICAM1 (Intercellular Adhesion Molecule 1) contributes to hepatic injury. In experimental models, modulation of Icam1 influenced adipose homeostasis and liver damage in the context of obesity, highlighting organ-specific functions relevant to metabolic liver disease [37]. Notably, functional polymorphisms in ICAM1, including rs5491 (p.K56M), exhibit striking differences in allele frequency between Black and White individuals, suggesting a potential genetic basis for racial disparities in liver disease susceptibility [38].

The interferon-gamma receptor 2 (IFNGR2) gene encodes a component of the type II interferon receptor complex, which is critical for mediating inflammatory and fibrotic pathways. Signaling through IFNGR2 has been implicated in liver inflammation and fibrosis in experimental steatohepatitis models [39].

The phosphoenolpyruvate carboxykinase 1 (PCK1) gene encodes a key gluconeogenic enzyme that also influences hepatic lipid metabolism. Experimental studies show that hepatic PCK1 deficiency promotes steatosis, inflammation, and fibrosis in mouse models, highlighting its role in pathways central to MASLD development and progression [40].

The PPP1R1A gene encodes protein phosphatase 1 regulatory inhibitor subunit 1A. Emerging evidence suggests that PPP1R1A may contribute indirectly to MASLD pathophysiology and racial differences in disease progression through dysregulation of hepatic glucose and lipid metabolism [41]. Transcriptomic analyses of hepatocytes from children with NAFLD show that PPP1R1A expression is reduced in diseased states compared with controls, supporting a potential role in hepatic metabolic responses and inflammation [42].

Thus, all race-specific MASLD-associated genes identified in our analysis have published evidence linking them to MASLD, further supporting the hypothesis that genes differentially expressed in the livers of Black and White individuals may contribute to racial disparities in MASLD. Gene network analysis indicates that these genes are largely independent, suggesting multiple distinct pathways may underlie these differences.

Gene set enrichment analysis did not identify any significantly enriched pathways or biological functions among genes differentially expressed between histologically normal livers of Black and White individuals. In contrast, genes associated with MASLD progression were significantly enriched for extracellular exosome–related genes, collagens, and endoplasmic reticulum–associated genes. These functional categories play critical roles in fibrogenesis [43–45], further supporting the conclusion that racial differences in liver gene expression profiles are more closely associated with MASLD progression rather than disease risk. Gene network analysis of race-specific MASLD-associated genes indicated that IFNGR2, CXCL9, and ICAM1 may directly interact with one another, whereas the remaining six genes were not involved in direct interactions and likely contribute independently to racial differences in MASLD progression.

The most significant gene differentially expressed between normal livers of Black and White individuals was SER-PINF1, making it a strong candidate MASLD risk-associated gene. SERPINF1 encodes pigment epithelium–derived factor (PEDF), a secreted glycoprotein belonging to the serpin family with well-documented anti-inflammatory, anti-oxidant, anti-angiogenic, and metabolic regulatory properties in hepatic and metabolic tissues. Human studies have shown that serum PEDF levels are elevated in individuals with steatosis and independently correlate with markers of liver fibrosis, such as procollagen type III N-terminal peptide, suggesting a link between PEDF and hepatic fibrogenic processes in

metabolic liver disease [46]. Mechanistic studies in experimental models further support a protective hepatic role for PEDF: in diet-induced steatohepatitis mouse models, adenoviral delivery of PEDF suppressed hepatic lipid accumulation, reduced oxidative stress, and downregulated pro-inflammatory cytokine expression [47]. In addition, PEDF has been shown to modulate triacylglycerol metabolism through interaction with adipose triglyceride lipase, thereby influencing lipid mobilization in both liver and adipose tissue [48]. Although direct human genetic association studies linking SERPINF1 to MASLD are currently limited, its function as a hepatokine/adipokine with pleiotropic effects on steatosis, immune signaling, and extracellular matrix remodeling makes SERPINF1 a biologically plausible candidate gene for MASLD risk.

## Limitations

We acknowledge that our study population was derived from a bariatric surgery cohort, which may limit direct generalizability to the broader population. Individuals undergoing bariatric surgery typically represent patients with more severe obesity and a higher burden of metabolic comorbidities, which may influence disease biology and gene expression patterns [49–51]. However, this relatively homogeneous population enables the detection of robust associations. While effect sizes may differ in non-obese populations, we expect the directionality and biological relevance of the findings to extend beyond this cohort. External validation in independent and more diverse populations will be important to confirm generalizability.

## Conclusion

In this study, we identified extensive race-associated differences in hepatic gene expression profiles and demonstrated that many of these genes overlap with the genes implicated in MASLD progression. Approximately one-third of analyzed MASLD-associated genes were differentially expressed between Black and White individuals. The substantial overlap between race-specific and MASLD-associated genes suggests that racial differences in liver gene expression may contribute to the observed disparities in MASLD susceptibility and progression.

Our findings underscore the importance of incorporating race as a biological and social variable in studies of MASLD pathogenesis and progression. Our analyses revealed novel candidate genes that were not previously detected in unstratified datasets, indicating that pooled analyses may obscure key molecular drivers of disease in specific populations. The increasing divergence in gene expression between racial groups with advancing disease stages further supports the need for race-aware approaches, particularly when investigating mechanisms of fibrosis and advanced MASLD. The widespread differences in gene expression may reflect both genetic and environmental factors, including race-specific genetic variants and dietary habits known to influence hepatic transcriptional regulation. Together, these findings highlight a complex interplay between genetic background, environmental exposures, and disease biology.

In conclusion, our study provides the first systematic characterization of race-specific hepatic gene expression signatures in MASLD. These data offer novel insights into the molecular underpinnings of racial disparities in MASLD and identify potential targets for precision prevention and treatment strategies. Future studies integrating genomic, epigenetic, and environmental data across diverse populations are warranted to validate these findings and to elucidate the mechanisms through which race-associated gene expression patterns contribute to MASLD development and progression.

The findings of this study should be interpreted as associative and hypothesis-generating rather than as evidence of causal or mechanistic differences. Although we observed race-associated variation in hepatic gene expression and enrichment of genes linked to MASLD progression, the observational nature of the analysis precludes direct inference of causality. These results instead provide a framework for future mechanistic studies aimed at elucidating the biological pathways through which race-associated molecular differences may contribute to MASLD susceptibility and progression.

## Supporting information

**S1 Table. Genes differentially expressed between Blacks and Whites at different stages of liver disease.**
(XLS)

**S2 Table. A comparison of mean gene expressions between different MASLD stages in Blacks and Whites.** Multiple Excel sheets.
(XLS)

**S3 Table. Gene expression significantly associated with liver disease stage after adjustment for multiple testing.**
(XLS)

**S4 Table. Genes significantly associated with race or MASLD stage based on two-way ANOVA.**
(XLS)

## Acknowledgments

The authors sincerely thank Mrs. Koshka Kakashka for help with statistical analysis.

## Author contributions

**Conceptualization:** Ivan P. Gorlov, Olga Y. Gorlova, Aaron P. Thrift.

**Data curation:** Ivan P. Gorlov.

**Formal analysis:** Ivan P. Gorlov.

**Investigation:** Olga Y. Gorlova, Aaron P. Thrift.

**Methodology:** Olga Y. Gorlova, Aaron P. Thrift.

**Resources:** Aaron P. Thrift.

**Validation:** Olga Y. Gorlova.

**Writing – original draft:** Ivan P. Gorlov.

**Writing – review & editing:** Olga Y. Gorlova, Aaron P. Thrift.

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
