## [Decision Letter · Decision Letter 0]

22 Dec 2025

PONE-D-25-58137Differences in Gene Expression May Contribute to the Racial Differences in the Risk of MASLDPLOS One

Dear Dr. Gorlov,

Thank you for submitting your manuscript to PLOS ONE. We have completed the review of your manuscript, and a summary containing the reviewers’ comments is appended below.  After careful consideration, we feel that it has merit but does not fully meet PLOS ONE’s publication criteria as it currently stands. Based on the reviewers' comments, a **major revision** is required.is required.

Both reviewers have expressed concerns about the work and the manuscript, specifically highlighting the limitations attributed to the study design, its statistical justification, data interpretation, and the biological relevance of the reported findings. These issues will need to be addressed with additional data or experiments provided in a successfully amended manuscript. A more rigorous statistical justification, along with a comprehensive and critical data discussion, is anticipated in the revised manuscript, leading to stronger, biologically relevant conclusions. Your revised manuscript will be re-evaluated by one or more original reviewers.

We invite you to submit a revised version of the manuscript that addresses the points raised during the review process. We look forward to receiving your point-by-point response to the reviewers and your revised manuscript.

If applicable, we recommend that you deposit your laboratory protocols in protocols.io to enhance the reproducibility of your results. Protocols.io assigns your protocol its own identifier (DOI) so that it can be cited independently in the future. For instructions see: https://journals.plos.org/plosone/s/submission-guidelines#loc-laboratory-protocols. Additionally, PLOS ONE offers an option for publishing peer-reviewed Lab Protocol articles, which describe protocols hosted on protocols.io. Read more information on sharing protocols at . Additionally, PLOS ONE offers an option for publishing peer-reviewed Lab Protocol articles, which describe protocols hosted on protocols.io. Read more information on sharing protocols at https://plos.org/protocols?utm_medium=editorial-email&utm_source=authorletters&utm_campaign=protocols..

We look forward to receiving your revised manuscript.

Kind regards,

Igor Shmarakov, Ph.D., Sc.D.

Academic Editor

PLOS One

Reviewers' comments:

Reviewer's Responses to Questions

**Comments to the Author**

1. Is the manuscript technically sound, and do the data support the conclusions?

Reviewer #1: Partly

Reviewer #2: Partly

2. Has the statistical analysis been performed appropriately and rigorously? 

Reviewer #1: Yes

Reviewer #2: Yes

3. Have the authors made all data underlying the findings in their manuscript fully available?

Reviewer #1: Yes

Reviewer #2: Yes

4. Is the manuscript presented in an intelligible fashion and written in standard English?

Reviewer #1: Yes

Reviewer #2: Yes

5. Review Comments to the Author

Reviewer #1: The paper by Dr. Gorlov et al aims to investigate differences in the gene expression of genes increased or decreased in MASLD and MASH comparing cohorts of obese White and Black individuals who underwent bariatric surgery. By re-analyzing the data from a previous study (Subudhi et al), the authors focus on the stratification between White and Black individuals to find potential genes predictive of progression to MASH.

While this study addresses important differences to keep into account when analyzing different populations, such as the differences in specific genes driving disease progression, it fails to fully conclude what those genes found signify in that context.

The paper published by Rich et al cited in this manuscript indicates that the highest risk for MASH is in Hispanic populations and lowest in Black populations vs White ones. So, while differences between Black and White populations can exist, these could be much smaller. One may ask how the genes that Dr. Gorlov et al found comparing the two populations relate to the Hispanic one, given that this is the population with the highest MAFLD burden.

It would be useful to know if the two cohorts analyzed in the manuscript, White and Black individuals, include people living in rural vs urban areas because access to food and its quality can be significantly different.

In Figure 1, please state if the age and BMI are statistically different or not. It seems that there are no statistical differences in these populations, but this needs to be clearly stated.

Are liver biopsies available for some of the cases analyzed? Integrating analyses of liver biopsies with the genes identified as the most different between the two cohorts, may lead to stronger conclusions and a better interpretation of the data. In fact, one of the major limitations of this study is the full interpretation of the role of these genes including MMP15, LAMB2, UCN3, PRSS3 among others in White vs Black populations. The interpretation of these data is superficial in the Results and Discussion sections and needs more strengthening. For example, the 9 genes identified in the Results section: Race-specific MASLD-associated genes, shows the list of the genes that result positively or negatively correlated with MASLD, but fails to frame these results in the context of the cases analyzed. As an example, COL5A2 increases correlate with MASLD in White individuals, whereas CXCL9 shows increases in White individuals but no changes in Black individuals and so on... Based on the results of these 9 genes, are White individuals with MASLD more predisposed to progression of disease than Black individuals or not? What do the pathways associated with the genes identified by the authors suggest for the 2 populations?

Extra point to increase the quality and clarity of the study:

The authors should strengthen the data by providing a conclusive sentence stating what each paragraph in the Results section mean. As it is now, the manuscript seems to present the data as a list of findings rather than a curated analysis leading us to learn the significance of the results.

Reviewer #2: This study reanalyzed publicly available NanoString nCounter gene expression data (GSE163211) from liver biopsies of 98 Black and 211 White individuals undergoing bariatric surgery. The authors have identified race-related differences in hepatic gene expression and how these differences relate to histologically defined stages of metabolic dysfunction-associated steatotic liver disease (MASLD) progression from normal liver, steatosis to MASH without or without fibrosis. Using t-tests, Spearman correlations, two-way ANOVA, FDR control, and pathway/network analyses, the authors report a substantial number of genes differ between racial groups. Consistent with the original study by Subudhi et al., the number of differentially expressed genes increases with more advanced disease stages. Interestingly, the findings indicate that several genes may be race-specific MASLD-associated. The manuscript is well written and provides detailed methodological descriptions; however, several issues related to statistical rigor, threshold justification, interpretation, and limitations require clarification before the conclusions can be fully supported.

- The authors analyze 800 preselected genes using an FDR < 0.1 threshold. While FDR correction is appropriate for controlling false discoveries, the choice of the 10% cutoff is not explained, which differs from the original study’s p < 0.01 threshold by Subudhi et al. This raises concerns about whether the cutoff may influence the number of reported race-specific or progression-associated genes, especially when findings are marginal. A brief justification for selecting FDR < 0.1 is recommended, along with sensitivity analyses (e.g., FDR < 0.05 or adjusted p-values from t-tests/ANOVA) to validate the main conclusions.

- Although the authors acknowledge the substantial imbalance in sample size between Group 4 (13 Black vs. 68 White participants), this disparity likely limits the power and stability of race-by-stage comparisons. The permutation analysis is helpful but does not fully address potential issues such as unstable variance estimates or unreliable interaction testing. A short discussion of how this imbalance affects interpretation, and the inclusion of effect sizes or confidence intervals for key findings, would strengthen the results.

- The interpretation of race-specific MASLD-associated genes might be somewhat overstated beyond what the data can support, given the cross-sectional design, lack of covariate adjustment, and absence of functional validation. It would strengthen the manuscript to frame these findings as associative and hypothesis-generating rather than implying causal or mechanistic differences.

- Although the original study discussed several demographic and metabolic covariates, the current re-analysis mainly uses unadjusted t-tests and ANOVA. Without including variables such as age, sex distribution, BMI, and metabolic comorbidities in the statistical models, it is hard to determine whether observed race-associated differences reflect biology or underlying group differences. A brief discussion of this limitation, or inclusion of covariate-adjusted analyses if the metadata permit, would improve interpretability.

- The manuscript should clarify whether the 800-gene panel was predetermined, even if this has been reported in the original study, or selected for this reanalysis. Because the analysis focuses on a constrained gene set rather than the full transcriptome, potential selection bias and missed pathways should be briefly acknowledged. A short explanation of how and why this panel was chosen would help readers interpret the scope and limitations of the findings.

6. PLOS authors have the option to publish the peer review history of their article (what does this mean?). If published, this will include your full peer review and any attached files.). If published, this will include your full peer review and any attached files.

.

Reviewer #1: No

Reviewer #2: No

---

## [Author Response · Author response to Decision Letter 1]

8 Jan 2026

Dear Dr. Shmarakov,

Thank you for the opportunity to revise and resubmit our manuscript entitled “Differences in Gene Expression May Contribute to the Racial Differences in the Risk of MASLD.” We sincerely appreciate the time and effort you and the two anonymous reviewers devoted to providing thoughtful and constructive feedback.

In response to the reviewers’ comments and suggestions, we have substantially revised and expanded the manuscript, with particular emphasis on strengthening and clarifying the Discussion section. We believe that these revisions have significantly improved the clarity, rigor, and overall impact of the work.

Below, we provide a detailed, point-by-point response to each reviewer comment, describing how each concern was addressed in the revised manuscript. We have uploaded two versions of the manuscript: a clean version and a version with highlighted changes for ease of review.

On behalf of all authors,

Ivan Gorlov

Reviewer 1.

While this study addresses important differences to keep into account when analyzing different populations, such as the differences in specific genes driving disease progression, it fails to fully conclude what those genes found signify in that context.

Our goal was less ambitious than “to fully conclude what those genes found signify in that context.” We want to bring attention to the possibility that genetic variation may contribute to ethnic differences in MASLD incidence. The typical explanation of ethnic differences in MASLD refers to the differences in lifestyle. This explanation is likely insufficient, especially when we consider Black population with higher prevalence of lifestyle risk factors and lower MASLD prevalence compared to Whites.

The paper published by Rich et al cited in this manuscript indicates that the highest risk for MASH is in Hispanic populations and lowest in Black populations vs White ones. So, while differences between Black and White populations can exist, these could be much smaller. One may ask how the genes that Dr. Gorlov et al found comparing the two populations relate to the Hispanic one, given that this is the population with the highest MAFLD burden.

There is some variation in estimates of MASLD prevalence in different populations. Meta-analysis suggests roughly 24-29% prevalence in Hispanics in the U.S. (PMID: 28970148) which is higher but still very similar to Non-Hispanic Whites (PMID: 28970148). Estimated prevalence in Blacks is ~12% (PMID: 23512725). Even though the estimated risk depends on method of diagnostic (ultrasound versus serum liver enzymes) and targeted population there is a general agreement that risk is high in U.S. Hispanics and Whites and low in African Americans. We did not include analysis of Hispanics as a separate group because population size was low.

It would be useful to know if the two cohorts analyzed in the manuscript, White and Black individuals, include people living in rural vs urban areas because access to food and its quality can be significantly different.

This is a good point, but unfortunately, we do not have these data.

In Figure 1, please state if the age and BMI are statistically different or not. It seems that there are no statistical differences in these populations, but this needs to be clearly stated.

In the revised manuscript, we provided statistics for age and BMI differences between Blacks and Whites. In both cases, the differences did not reach the level of statistical significance: p=0.09 for age and p=0.25 for BMI.

Are liver biopsies available for some of the cases analyzed? Integrating analyses of liver biopsies with the genes identified as the most different between the two cohorts, may lead to stronger conclusions and a better interpretation of the data.

No, liver biopsies are not available.

In fact, one of the major limitations of this study is the full interpretation of the role of these genes including MMP15, LAMB2, UCN3, PRSS3 among others in White vs Black populations. The interpretation of these data is superficial in the Results and Discussion sections and needs more strengthening. For example, the 9 genes identified in the Results section: Race-specific MASLD-associated genes, shows the list of the genes that result positively or negatively correlated with MASLD, but fails to frame these results in the context of the cases analyzed. As an example, COL5A2 increases correlate with MASLD in White individuals, whereas CXCL9 shows increases in White individuals but no changes in Black individuals and so on... Based on the results of these 9 genes, are White individuals with MASLD more predisposed to progression of disease than Black individuals or not? What do the pathways associated with the genes identified by the authors suggest for the 2 populations?

We substantially extended the Introduction section to justify the need of comparative genetic analysis of MASLD risk and progression for a better understanding of factors contributing to differences between Black and White individuals. We extended the Results section and significantly extended the Discussion section to discuss identified candidate genes contributing to racial differences for MASLD risk and progression. We added a paragraph on pathway analysis of the identified candidates.

Reviewer 2.

This study reanalyzed publicly available NanoString nCounter gene expression data (GSE163211) from liver biopsies of 98 Black and 211 White individuals undergoing bariatric surgery. The authors have identified race-related differences in hepatic gene expression and how these differences relate to histologically defined stages of metabolic dysfunction-associated steatotic liver disease (MASLD) progression from normal liver, steatosis to MASH without or without fibrosis. Using t-tests, Spearman correlations, two-way ANOVA, FDR control, and pathway/network analyses, the authors report a substantial number of genes differ between racial groups. Consistent with the original study by Subudhi et al., the number of differentially expressed genes increases with more advanced disease stages. Interestingly, the findings indicate that several genes may be race-specific MASLD-associated. The manuscript is well written and provides detailed methodological descriptions; however, several issues related to statistical rigor, threshold justification, interpretation, and limitations require clarification before the conclusions can be fully supported. The authors analyze 800 preselected genes using an FDR < 0.1 threshold. While FDR correction is appropriate for controlling false discoveries, the choice of the 10% cutoff is not explained, which differs from the original study’s p < 0.01 threshold by Subudhi et al. This raises concerns about whether the cutoff may influence the number of reported race-specific or progression-associated genes, especially when findings are marginal. A brief justification for selecting FDR < 0.1 is recommended, along with sensitivity analyses (e.g., FDR < 0.05 or adjusted p-values from t-tests/ANOVA) to validate the main conclusions.

We used FDR threshold of 0.1 because our analysis is exploratory and aims to identify and prioritize genes potentially meaningful for racial differences in MASLD risk and progression. Controlling the false discovery rate at 10% allows adequate sensitivity to detect true effects that may be modest in size, which is particularly important when multiple testing reduces statistical power. An FDR of 0.1 is widely accepted in discovery-oriented studies and provides a reasonable trade-off between identifying relevant candidates and limiting the expected proportion of false positives among the reported findings.

- Although the authors acknowledge the substantial imbalance in sample size between Group 4 (13 Black vs. 68 White participants), this disparity likely limits the power and stability of race-by-stage comparisons. The permutation analysis is helpful but does not fully address potential issues such as unstable variance estimates or unreliable interaction testing. A short discussion of how this imbalance affects interpretation, and the inclusion of effect sizes or confidence intervals for key findings, would strengthen the results.

The following paragraph was added: “The smaller number of DE genes observed in Group 4 compared with Group 3 is likely attributable to the markedly unbalanced sample size in Group 4 (68 White vs. 13 Black individuals). Because statistical power is largely driven by the smaller group, this imbalance results in wider confidence intervals and reduced sensitivity to detect true expression differences. In addition, smaller sample sizes are more susceptible to outliers and random variability, further limiting detection power. Consequently, a direct comparison of p-values across groups with balanced and unbalanced sample sizes may be misleading.”

The interpretation of race-specific MASLD-associated genes might be somewhat overstated beyond what the data can support, given the cross-sectional design, lack of covariate adjustment, and absence of functional validation. It would strengthen the manuscript to frame these findings as associative and hypothesis-generating rather than implying causal or mechanistic differences.

The following paragraph was included at the end of the Discussion section: “The findings of this study should be interpreted as associative and hypothesis-generating rather than as evidence of causal or mechanistic differences. Although we observed race-associated variation in hepatic gene expression and enrichment of genes linked to MASLD progression, the observational nature of the analysis precludes direct inference of causality. These results instead provide a framework for future mechanistic studies aimed at elucidating the biological pathways through which race-associated molecular differences may contribute to MASLD susceptibility and progression.”

Although the original study discussed several demographic and metabolic covariates, the current re-analysis mainly uses unadjusted t-tests and ANOVA. Without including variables such as age, sex distribution, BMI, and metabolic comorbidities in the statistical models, it is hard to determine whether observed race-associated differences reflect biology or underlying group differences. A brief discussion of this limitation, or inclusion of covariate-adjusted analyses if the metadata permit, would improve interpretability.

Black and White individuals included in this analysis did not differ significantly in age or BMI, and the sex composition was also comparable between the groups. Because individual-level data on clinical comorbidities were not available, we were unable to assess differences in comorbidity burden between groups, which represents a limitation of this study. Accordingly, the following paragraph was added to the revised manuscript: “Although the compared groups did not differ by age, BMI, or sex composition, we were unable to compare them with respect to clinical comorbidities due to the lack of individual-level data. This limitation should be considered when interpreting the results of this study.”

The manuscript should clarify whether the 800-gene panel was predetermined, even if this has been reported in the original study, or selected for this reanalysis. Because the analysis focuses on a constrained gene set rather than the full transcriptome, potential selection bias and missed pathways should be briefly acknowledged. A short explanation of how and why this panel was chosen would help readers interpret the scope and limitations of the findings.

The following text was added to the revised manuscript: “Gene expression of 800 genes was analyzed. These genes were selected based on published evidence supporting their roles in liver disease and fibrosis, reflecting a candidate gene approach. To account for the preselected nature of this gene set and to minimize potential selection bias, we used these 800 genes rather than the entire human genome as the background for gene set enrichment analysis.”

---

## [Decision Letter · Decision Letter 1]

1 Apr 2026

PONE-D-25-58137R1Differences in Gene Expression May Contribute to the Racial Differences in the Risk of MASLDPLOS One

Dear Dr. Gorlov,

Thank you for submitting your manuscript to PLOS ONE. After careful consideration, we feel that it has merit but does not fully meet PLOS ONE’s publication criteria as it currently stands. Therefore, we invite you to submit a revised version of the manuscript that addresses the points raised during the review process. 

If applicable, we recommend that you deposit your laboratory protocols in protocols.io to enhance the reproducibility of your results. Protocols.io assigns your protocol its own identifier (DOI) so that it can be cited independently in the future. For instructions see: https://journals.plos.org/plosone/s/submission-guidelines#loc-laboratory-protocols. Additionally, PLOS ONE offers an option for publishing peer-reviewed Lab Protocol articles, which describe protocols hosted on protocols.io. Read more information on sharing protocols at . Additionally, PLOS ONE offers an option for publishing peer-reviewed Lab Protocol articles, which describe protocols hosted on protocols.io. Read more information on sharing protocols at https://plos.org/protocols?utm_medium=editorial-email&utm_source=authorletters&utm_campaign=protocols..

As the corresponding author, your ORCID iD is verified in the submission system and will appear in the published article. PLOS supports the use of ORCID, and we encourage all coauthors to register for an ORCID iD and use it as well. Please encourage your coauthors to verify their ORCID iD within the submission system before final acceptance, as unverified ORCID iDs will not appear in the published article. *Only* the individual author can complete the verification step; PLOS staff the individual author can complete the verification step; PLOS staff *cannot* verify ORCID iDs on behalf of authors.verify ORCID iDs on behalf of authors.

We look forward to receiving your revised manuscript.

Kind regards,

Nobuyuki Takahashi, Ph.D.

Academic Editor

PLOS One

Journal Requirements:

Additional Editor Comments:

The reviewers remained concerned about the strength and clarity of the analysis. Therefore, we invited an additional reviewer who specialises in analytical methods. Please consider the additional comments and address them.

I hope you understand the potentially sensitive nature of this study on racial differences in the risk of a disease.

Reviewers' comments:

Reviewer's Responses to Questions

**Comments to the Author**

1. If the authors have adequately addressed your comments raised in a previous round of review and you feel that this manuscript is now acceptable for publication, you may indicate that here to bypass the “Comments to the Author” section, enter your conflict of interest statement in the “Confidential to Editor” section, and submit your "Accept" recommendation.

Reviewer #1: All comments have been addressed

Reviewer #2: All comments have been addressed

Reviewer #3: (No Response)

2. Is the manuscript technically sound, and do the data support the conclusions?

Reviewer #1: Yes

Reviewer #2: Yes

Reviewer #3: (No Response)

3. Has the statistical analysis been performed appropriately and rigorously? 

Reviewer #1: Yes

Reviewer #2: Yes

Reviewer #3: (No Response)

4. Have the authors made all data underlying the findings in their manuscript fully available?

Reviewer #1: Yes

Reviewer #2: Yes

Reviewer #3: (No Response)

5. Is the manuscript presented in an intelligible fashion and written in standard English?

Reviewer #1: Yes

Reviewer #2: Yes

Reviewer #3: (No Response)

6. Review Comments to the Author

Reviewer #1: The authors addressed most of the reviewer's concerns and the manuscript's quality is improved now.

Reviewer #2: (No Response)

Reviewer #3: The research team recruited 300 Black and White individuals with bariatric surgery to test whether race-associated variation in hepatic gene expression may contribute to differential progression of MASLD. They concluded that differential modulation of hepatic gene expression may contribute to racial disparities in MASLD.

1. 800 genes were analyzed based on published evidence. Please cite the relevant reference to support the selection.

2. The language in the manuscript wasn’t precise and may be misleading. For example, In the results section, (a) it stated no significant differences for age and bmi while it also stated “slightly higher” and “marginally higher”. They conflict with each other; (2) the number of DE genes increased with disease severity, with 41, 141, 316, and 136 ,…, it decreases from group 3 to group 4!

3. The authors suspected that the smaller number of DE genes in Group 4 compared with Group 3 is due to an unbalanced sample. Can this be checked directly based on the data to provide further support for this suspicion? Alternatively, the counts can be reported by ancestry/race.

4. Genes differentially expressed in stage transition were reported. Please clarify the study design on the timeline for gene expression assessment, and stage information and its transition. How to obtain the transition of the stage?

5. In the joint analysis of race and stage effects. It’s unclear about the reported associations. For example, authors reported 232 genes associated with race. As the model contains the interaction, it’s clear it refers to the main effect for this association or the joint effect of the main and interaction effect.

6. Preselected genes may limit discovery space and bias the downstream pathway enrichment analysis as well. Are the pathway enrichment analysis results from the preselected 800 genes different from the results from refined gene list?

7. MASLD stage was treated as an ordinal numeric variable for correlation calculation. It might be more suitable to conduct stage-wise contrast instead. Have authors considered conducting ordinal regression?

8. It would be informative to formally test the race x stage interactions.

9. The sample was based on a bariatric surgery cohort. Please comment on the generalizability of the results.

7. PLOS authors have the option to publish the peer review history of their article (what does this mean?). If published, this will include your full peer review and any attached files.). If published, this will include your full peer review and any attached files.

.

Reviewer #1: No

Reviewer #2: No

Reviewer #3: No

---

## [Author Response · Author response to Decision Letter 2]

7 Apr 2026

First two reviewers were satisfied with our responses and do not have additional comments and suggestions. In response to the reviewers’ 3 comments and suggestions, we conducted additional analyses and have substantially revised the manuscript to improve the clarity and rigor of the study.

Below, we provide a detailed, point-by-point response to reviewer 3 comment and describe how the concern was addressed in the revised manuscript. We have uploaded two versions of the manuscript: a clean version and a version with highlighted changes for ease of review.

Reviewer 1.

Reviewer 1 was satisfied with our responses and did not have any additional comments and suggestions.

Reviewer 2.

Reviewer 2 was satisfied with our responses and did not have any additional comments and suggestions.

Reviewer 3.

1. 800 genes were analyzed based on published evidence. Please cite the relevant reference to support the selection.

We have now included three references to justify the selection of MASLD-associated genes.

2. The language in the manuscript wasn’t precise and may be misleading. For example, In the results section, (a) it stated no significant differences for age and bmi while it also stated “slightly higher” and “marginally higher”. They conflict with each other; (2) the number of DE genes increased with disease severity, with 41, 141, 316, and 136 ,…, it decreases from group 3 to group 4!

We apologize for the lack of clarity. We agree that phrases such as “slightly higher” or “marginally higher” may be misleading when differences are not statistically significant. We have revised the text to remove such wording and present the results more formally:

“The average age in White individuals was 45.2 ± 1.1 years and in Black individuals 42.4 ± 1.5 years (t = 1.6, p = 0.09). BMI in Black individuals was 48.0 ± 1.1 kg/m² and in White individuals 46.2 ± 0.7 kg/m² (t = 1.2, p = 0.25).”

3. The authors suspected that the smaller number of DE genes in Group 4 compared with Group 3 is due to an unbalanced sample. Can this be checked directly based on the data to provide further support for this suspicion? Alternatively, the counts can be reported by ancestry/race.

To address this concern, we conducted additional analyses to evaluate the impact of sample size imbalance. Specifically, in Group 3 (“NASH without fibrosis”), we performed permutation analyses by randomly selecting 13 out of 22 Black individuals and comparing them with 49 White individuals. Across 50 permutations, the average number of DE genes was 108.5 ± 4.3, substantially lower than the 316 DE genes identified in the full dataset.

Analysis of mean absolute differences in gene expression between Black and White individuals across consecutive MASLD stages further supports the conclusion that Black versus White divergence in gene expression profiles increases with disease progression. The mean absolute log ratios of gene expression between Black and White individuals were 0.103 ± 0.005 for normal liver histology, 0.121 ± 0.008 for steatotic liver, 0.133 ± 0.005 for NASH without fibrosis, and 0.161 ± 0.007 for NASH with fibrosis.

4. Genes differentially expressed in stage transition were reported. Please clarify the study design on the timeline for gene expression assessment, and stage information and its transition. How to obtain the transition of the stage?

We clarify stage transition analysis in the revised manuscript. The following text was added to the Results section: “Three consecutive transitions associated with MASLD progression were analyzed: (1) the transition from a liver with normal histology to steatotic liver, (2) the transition from steatotic liver to MASH without fibrosis, and (3) the transition from MASH without fibrosis to MASH with fibrosis. We wanted to identify genes differentially expressed between MASLD consecutive stages. We ran the joint analysis with both races together and race-stratified analyses. In the analysis of “all races together”, no differentially expressed genes were detected for the first transition.”

5. In the joint analysis of race and stage effects. It’s unclear about the reported associations. For example, authors reported 232 genes associated with race. As the model contains the interaction, it’s clear it refers to the main effect for this association or the joint effect of the main and interaction effect.

We provide the results for main effects (race and stage) and the “race x stage” interaction term. We clarified this in the revision: “We used a two-way ANOVA to estimate the effects of race and stage (categorical variable) on gene expression levels (continuous dependent variable), including a race × stage interaction term in the model. We identified 232 genes for which the main effect of race remained statistically significant in the presence of the race x stage interaction term after adjustment for multiple testing (p < 0.05). In contrast, the main effect of stage remained statistically significant for only 56 genes after multiple testing correction. Twenty-six genes were significantly associated with both race and stage, which is comparable to the expected number (16.2) of genes significant for both factors under independence. The results of this analysis are presented in Supplementary Table S4.”

6. Preselected genes may limit discovery space and bias the downstream pathway enrichment analysis as well. Are the pathway enrichment analysis results from the preselected 800 genes different from the results from refined gene list?

We acknowledge that restricting the analyses to preselected genes may introduce bias. To reiterate, our choice was limited to these genes, since our paper is a secondary analysis of an existing dataset that included these preselected 800 genes. To mitigate selection bias, we used the same preselected gene set as the background for pathway enrichment analyses. This rationale is emphasized in the revised manuscript.

7. MASLD stage was treated as an ordinal numeric variable for correlation calculation. It might be more suitable to conduct stage-wise contrast instead. Have authors considered conducting ordinal regression?

We thank the reviewer for this important comment. We agree that treating MASLD stage as a continuous numeric variable may not fully capture its ordinal nature. We performed additional analyses to better account for the ordered categorical structure of the outcome.

Following the Reviewer recommendation, we implemented ordinal regression (proportional odds model), which models the ordered nature of MASLD stage. The ordinal regression was run using joint sample (Blacks and Whites together) and separately on Black and Whites individuals. The results of the analyses are shown on 3 additional sheets in supplementary table S3. A total of 76 genes were identified as stage associated in at least one type of analysis. Most of the genes, 73 out of 76, were detected by Spearman’s rank correlation. Three genes: CFB, PKLR and UBL5 were detected by ordinal regression but not by correlation analysis. Therefore, the results of ordinal regression analysis are similar and complementary to the results generated by rank correlation and identified genes whose expression level correlates with MASLD progression.

8. It would be informative to formally test the race x stage interactions.

Additional supplementary table S4 was included in the revised paper. The table shows the F statistics and p-values for “race”, “stage”, and interaction term “race x stage” derived from two-way ANOVA analysis.

9. The sample was based on a bariatric surgery cohort. Please comment on the generalizability of the results.

We have added a “Limitations” section addressing generalizability:

“We acknowledge that our study population was derived from a bariatric surgery cohort, which may limit direct generalizability to the broader population. Individuals undergoing bariatric surgery typically represent patients with more severe obesity and a higher burden of metabolic comorbidities, which may influence disease biology and gene expression patterns (PMID: 41172445, 39920373, 36528638). However, this relatively homogeneous population enables the detection of robust associations. While effect sizes may differ in non-obese populations, we expect the directionality and biological relevance of the findings to extend beyond this cohort. External validation in independent and more diverse populations will be important to confirm generalizability.”

---

## [Decision Letter · Decision Letter 2]

15 Apr 2026

Differences in Gene Expression May Contribute to the Racial Differences in the Risk of MASLD

PONE-D-25-58137R2

Dear Dr. Gorlov,

We’re pleased to inform you that your manuscript has been judged scientifically suitable for publication and will be formally accepted for publication once it meets all outstanding technical requirements.

An invoice will be generated when your article is formally accepted. Please note, if your institution has a publishing partnership with PLOS and your article meets the relevant criteria, all or part of your publication costs will be covered. Please make sure your user information is up-to-date by logging into Editorial Manager at Editorial Manager® and clicking the ‘Update My Information' link at the top of the page. For questions related to billing, please contact  and clicking the ‘Update My Information' link at the top of the page. For questions related to billing, please contact billing support..

Kind regards,

Nobuyuki Takahashi, Ph.D.

Academic Editor

PLOS One

Reviewers' comments:

Reviewer's Responses to Questions

**Comments to the Author**

1. If the authors have adequately addressed your comments raised in a previous round of review and you feel that this manuscript is now acceptable for publication, you may indicate that here to bypass the “Comments to the Author” section, enter your conflict of interest statement in the “Confidential to Editor” section, and submit your "Accept" recommendation.

Reviewer #3: All comments have been addressed

2. Is the manuscript technically sound, and do the data support the conclusions?

Reviewer #3: (No Response)

3. Has the statistical analysis been performed appropriately and rigorously? 

Reviewer #3: (No Response)

4. Have the authors made all data underlying the findings in their manuscript fully available?

Reviewer #3: (No Response)

5. Is the manuscript presented in an intelligible fashion and written in standard English?

Reviewer #3: (No Response)

6. Review Comments to the Author

Reviewer #3: (No Response)

7. PLOS authors have the option to publish the peer review history of their article (what does this mean?). If published, this will include your full peer review and any attached files.). If published, this will include your full peer review and any attached files.

.

Reviewer #3: No

---

## [Editor Report · Acceptance letter]

PONE-D-25-58137R2

PLOS One

Dear Dr. Gorlov,

I'm pleased to inform you that your manuscript has been deemed suitable for publication in PLOS One. Congratulations! Your manuscript is now being handed over to our production team.

Kind regards,

on behalf of

Dr. Nobuyuki Takahashi

Academic Editor

PLOS One